# Association between Electronic Media Use and Internalizing Problems: The Mediating Effect of Parent–Child Conflict and Moderating Effect of Children’s Age

**DOI:** 10.3390/bs13080694

**Published:** 2023-08-21

**Authors:** Shuliang Geng, Ke Xu, Xiaocen Liu

**Affiliations:** College of Preschool Education, Capital Normal University, Beijing 100048, China; 2223102004@cnu.edu.cn (S.G.); 18336930031@163.com (K.X.)

**Keywords:** children, electronic media, internalizing problems, parent–child conflict, age, anxiety, depression, social withdrawal

## Abstract

In today’s digital world, children are exposed extensively to electronic media, making it an integral part of their daily lives. However, excessive use of electronic media during childhood has been associated with various internalizing problems. Moreover, parent–child conflict and children’s age may be closely associated with children’s problem behaviors. The current study employed a cross-sectional design and conducted a questionnaire survey of 711 parents to examine the association between children’s electronic media use and their internalizing problems. Furthermore, this study probed the mediating role of parent–child conflict within this association and the moderating effect of children’s age. The results of structural equation modeling showed a positive correlation between children’s use of electronic media and their internalizing difficulties. Parent–child conflict served as a mediating factor in this association. Results also showed that the association between parent–child conflict and internalizing problems becomes more pronounced as children grow older. These findings imply that parents should encourage their children to develop healthy habits in using electronic media while fostering positive relationships. Parents should also be mindful of the psychological changes as children age and provide guidance to help them become proficient digital citizens.

## 1. Introduction

Investigating problem behaviors in early childhood holds substantial academic significance because such behaviors profoundly influence the individual’s social adaptation and interpersonal engagements [1]. The prevalence of emotional issues among children in China is consistently increasing, displaying a gradually decreasing age of onset. Nonetheless, children’s emotional problems are mainly detected by clinicians because of limited parental attention and awareness of children’s internalization difficulties. Therefore, when internalizing problems are identified, irreversible damage has often occurred, jeopardizing the child’s future academic achievement and social development [2]. 

“Internalizing problems” refer to the internal issues that individuals manifest during social interactions, primarily in the form of emotional disturbances such as anxiety and depression, as well as difficulties in peer engagement such as withdrawal [3]. Insufficient cognitive capability in children leads to deficiencies in developing strategies for regulating emotions and coping with stress, rendering them susceptible to experiencing internalizing problems [4]. As a result of maladaptive parenting behaviors, such as hostility and punitive measures, some Chinese children often exhibit reduced autonomy, limited attentional capacities, and increased dependency, potentially elevating their susceptibility to depressive symptoms and social withdrawal [5]. Research conducted by Yuan et al. revealed a notably high prevalence of depression and anxiety among a sample of 1412 children and adolescents in Beijing, with rates reaching approximately 13.1% and 31.1%, respectively [6]. Furthermore, several studies have indicated that internalizing problems, especially depression, are frequently observed among Chinese children [7].

Children’s internalizing problems are intimately related to age. Around the age of 1 year, there is a high prevalence of postnatal depression and anxiety in mothers, which may trigger children’s internalizing problems [8]. Zhou et al. also highlighted the association between children’s perceived conflict and internalizing problems in their first two years of life [9]. Moreover, children in the pre-school period show relatively high anxiety levels, although children’s internalizing problems during this period tend to decrease with age [10]. However, the transition to primary school, changes in learning styles, and increased peer competition create particular challenges for children [11]. Adolescents are highly susceptible to internalizing problems such as anxiety and depression because they experience physiological and psychological shifts and an expansion in their social network [12]. Min et al. also identified that early adolescents tended to exhibit more substantial internalizing problems with social withdrawal due to the large number of adaptive challenges they face [13]. Therefore, examining internalizing problems in the broader age range of Chinese children is necessary.

### 1.1. Electronic Media Use and Internalizing Problems

“Electronic media use” is defined as an individual’s use of electronic devices such as cell phones and computers [14]. With the rapid development of information technology, electronic devices are gaining popularity. Simultaneously, electronic media’s visually captivating imagery and sonorous audio effortlessly captivate children’s attention, increasing their exposure to such media [15]. Consequently, the employment of electronic media is regarded as a risk factor for instigating internalizing issues in children [16]. Brunborg et al. found that children who exhibit excessive addiction to video games display heightened emotional sensitivity, rendering them more susceptible to experiencing depression and anxiety [17]. The immersion of children in virtual realms can supplant their engagement in real-life socialization, leading to a sense of detachment from interpersonal interactions and consequently amplifying the likelihood of social withdrawal in children [18].

**Hypothesis 1.** 
*Children with higher levels of electronic media use are likely to experience social adjustment difficulties*
*, i.e., children’s electronic media use positively correlates with internalizing problems.*


### 1.2. The Mediating Role of Parent–Child Conflict

The displacement hypothesis states that individuals possess a finite amount of time and that excessive utilization of electronic media consumes a substantial portion of an individual’s time, which might reduce the opportunity for children to have direct interpersonal interactions with their families [19]. Therefore, children’s extended exposure to electronic media may engender a sense of “expectation deviation” among parents. In other words, children’s overuse of electronic media damages healthy parent–child relationships because it contradicts parental expectations that their children will disengage from electronic devices and actively engage in face-to-face interactions [20]. Parent–child conflict is a psychological discord or outward behavior resulting from cognitive, emotional, and behavioral disparities, and is a manifestation of parent–child relationship disharmony [21]. For instance, Charlie et al. found a correlation between children’s exposure to electronic media and deterioration in parent–child relationships [22]. Moreover, research by Venkatesh et al. indicated that children’s problematic Internet usage might have an additional impact on their parents’ occupational achievements [23]. The over-exposure of children to electronic media has been shown to contribute to increased parental stress in juggling work and parenting responsibilities, resulting in a decline in the quality of the parent–child relationship [24].

Davies and Cummings posited an emotional security hypothesis, suggesting that insecure parent–child relationships are detrimental to developing secure parent–child attachments, which might lead to problematic behaviors in children [25]. Attachment theory further underscores the notion that insecure parent–child attachment impedes children’s personal development and subsequent social adaptation [26], potentially fostering tendencies towards intimacy avoidance and Internet addiction [27]. Lippold et al. found that decreased emotional warmth and increased hostility in the parent–child relationship were strong indicators of early internalizing problems in adolescents [28]. Consequently, it is plausible to postulate that parent–child conflict might mediate the relationship between children’s electronic media use and internalizing problems.

**Hypothesis 2.** 
*Parent–child conflict mediates the relationship between children’s electronic media use and internalizing problems. Specifically, children’s enhanced electronic media use is accompanied by increased parent–child conflict, leading to a rise in internalizing difficulties among the children.*


### 1.3. The Moderating Role of Children’s Age

Several variables, including the age of the children, may operate as moderators in the association between children’s electronic media use and internalization problems through parent–child conflict [29]. Ecological systems theory states that an active two-way interaction exists between children and their surrounding environment, with characteristics such as children’s age and personality also influencing parental behavior [30]. Therefore, parent–child conflict arising from children’s electronic media use may be moderated by temporal systems such as children’s age. The “family life stage development theory” also proposes that as children grow older, they undergo distinct and unique developmental changes, which subsequently influence the dynamics of the parent–child relationship [31]. As children mature, they gradually tend to make independent decisions and judgments, and become more resistant to parental intervention [32]. Some children may extend the negative feelings associated with parent–child conflict to other interpersonal relationships and experience social adjustment difficulties [33,34]. Parent–child conflict arising from children’s exposure to electronic media may be perceived by older children as a form of regulatory behavior, which is highly likely to enhance the individual’s rebelliousness [35]. Therefore, an increase in children’s rebelliousness is commonly linked to parent–child conflict, which often leads to an exacerbation of internalizing problems such as social withdrawal [36,37]. Additionally, as children progress from kindergarten to elementary school, they undergo a significant developmental transition. Children may experience high academic pressure, peer competition, and academic distress during this particular stage due to parental expectations [38]. Moreover, in a competitive environment, children frequently receive less emotional care, and this lack of concern regularly results in intensified parent–child conflict and serious emotional problems such as depression and anxiety [39].

**Hypothesis 3.** 
*As children grow older, they become more resistant to parental intervention, triggering more substantial adverse effects. In other words, children’s age amplifies the positive relationship between parent–child conflict and internalizing problems.*


In summary, a review of the literature indicates that children’s electronic media use positively relates to internalizing problems [16,17,18]. Additionally, children’s engagement with electronic media exhibits a positive association with parent–child conflict [20]. Greater levels of parent–child conflict are linked to an elevated likelihood of anxiety and social adjustment challenges in children [27,28]. Moreover, most previous research has predominantly conceptualized children as passive participants in the perception of parent–child conflict, prioritizing parental agency and highlighting the adverse consequences of conflict on children. Nevertheless, it is essential to acknowledge that children are not solely passive entities within the parent–child dynamic. Factors such as age, temperament, and other individual characteristics are also associated with the intricacies of the parent–child relationship. In other words, children’s age may moderate the relationship between parent–child conflict and children’s internalizing problems. The specific hypotheses of the current study are depicted in Figure 1.

## 2. Materials and Methods

### 2.1. Respondents

The respondents were from the northern part of China. The initial sample consisted of 796 parents. We did not consider 85 respondents in the subsequent analysis due to incomplete information and contradictory answers. A total of 711 responses were included in the final analysis. The age range of the respondents’ children was 1.08 to 13.09 years, with a mean age of 5.23 years (*SD* = 1.91). There were 342 (48.1%) boys and 369 (51.9%) girls among the children, including 557 (78.3%) only children and 154 (21.7%) non-only children. Regarding the location of the family, 533 (75.0%) children lived in cities and 178 (25.0%) in towns or rural areas. The demographic characteristics of the parents are shown in Table 1.

### 2.2. Measures

#### 2.2.1. Electronic Media Use

The Electronic Media Use Questionnaire was adapted from the Video Game Use Questionnaire by Huang et al. [40]. We changed the term “video games” to “electronic media” in order to better encompass a broader range of media genres. The initial questionnaire comprised 20 items. After performing a validated factor analysis with a sample of 277 children, the final questionnaire was reduced to 14 items in four dimensions. The questionnaire consisted of electronic media time management (e.g., “Your child spends more time using electronic media than before”), interpersonal and health conditions caused by electronic media use (e.g., “Your child often makes new friends through electronic media”), life conflicts arising from electronic media use (e.g., “Your child spends less time playing outdoors because of the use of electronic media”), and emotional experiences related to electronic media use (e.g., “Your child gets angry at you for limiting his/her time using electronic media”). The questionnaire was scored on a 5-point Likert-type scale, with 1 indicating “strongly disagree” and 5 indicating “strongly agree”. Children’s electronic media use scores were summed across all items. The children’s Electronic Media Use Questionnaire scores ranged from 14–70. The higher the total score, the more serious the electronic media usage. The Cronbach’s alpha coefficient of the questionnaire in this study was 0.93, and the coefficients of each sub-dimension were 0.73, 0.80, 0.77, and 0.82, respectively. The structural validity was χ^2^/df = 2.71, RMSEA = 0.08, GFI = 0.91, NFI = 0.91, IFI = 0.94, TFI = 0.92, and CFI = 0.94.

#### 2.2.2. Parent–Child Conflict

The study employed the Child–Parent Relationship Scale (CPRS), originally developed by Pianta and subsequently revised by Zhang et al. [41]. The scale includes two dimensions: closeness (e.g., “I share an affectionate, warm relationship with my child”) and conflict (e.g., “My child and I always seem to be struggling with each other”). The scale consists of 22 items and is scored on a 5-point Likert-type scale. A total of 12 items on the conflict dimension were used to calculate the parent–child conflict score. The cumulative count of items encompassing the parent–child conflict subscale was utilized to derive the parent–child conflict score. The parent–child conflict subscale scores spanned the range from 12 to 60. The higher the score, the more serious the conflict between parents and children. The scale has demonstrated strong reliability and validity in the context of research on parent–child relationships [42]. The Cronbach’s alpha coefficient of the parent–child conflict subscale was 0.91, reflecting the high internal consistency of the scale. The structural validity of the scale has been demonstrated to be χ^2^/df = 2.02, RMSEA = 0.05, GFI = 0.92, and CFI = 0.91 [43].

#### 2.2.3. Internalizing Problems

The study also utilized the Strengths and Difficulties Questionnaire (SDQ), initially formulated by Goodman [44] and subsequently revised by Aarø et al. [45]. Goodman et al. initially established the viability of employing the Strengths and Difficulties Questionnaire for evaluating mental health among children aged 5 to 16 through a combination of questionnaire-based assessments and clinical diagnostic cross-validation [46]. Building upon this foundation, subsequent investigations by Patel et al. and Maurice-Stam et al. further validated the questionnaire’s utility for other age groups, specifically children aged 12 to 24 months and 2 to 18 years, respectively [47,48]. This cumulative evidence underscores the efficacy of the Strengths and Difficulties Questionnaire in comprehensively gauging problematic behavioral patterns among children spanning the age range of 1 to 18 years. The scale comprises three dimensions: externalizing problems, internalizing problems, and prosocial behaviors. The scale contains 21 items and is scored on a 3-point scale, with 0 meaning “not true” and 2 meaning “certainly true”. Following the guidelines established by Aarø et al. [45], seven items were chosen to assess children’s internalizing problems. The summation of scores across these seven items yielded the children’s internalizing problems score. The internalizing problems subscale exhibits a scoring spectrum spanning from 0 to 14. The higher the score, the more serious the child’s internalizing problems. The scale has good reliability in studies with groups of children [45]. The internalizing problems subscale exhibited a Cronbach’s alpha coefficient of 0.77. The structural validity was assessed using various fit indices, including χ^2^/df = 2.43, RMSEA = 0.04, GFI = 0.95, NFI = 0.91, IFI = 0.94, TFI = 0.93, and CFI = 0.94.

### 2.3. Procedures

Data and consent forms for this research were obtained from the parents who resided with the respective children. To ensure clarity, especially for families with multiple children, we instructed parents to provide only information about a single child. Throughout the data-gathering phase, participants were offered detailed clarifications on study variables. For example, within the questionnaire’s guidance section, the term “electronic media” was defined as encompassing platforms such as the Internet, computers, tablets, smartphones, digital television, video games, artificial intelligence aides, and intelligent learning devices that harness electronic, computer, and Internet-based technologies.

### 2.4. Statistical Analysis

Initial analyses of the collected data, including data cleansing, descriptive statistics, and correlation analysis, were performed using IBM SPSS software version 23.0. Subsequently, a moderated mediated model analysis was conducted with the assistance of the IBM SPSS Amos program version 26.0, and the moderating role of children’s age was further explored through a simple slope analysis.

## 3. Results

### 3.1. Common Method Bias

Following the suggestions of Podsakoff et al., Harman’s single-factor test was used to test for common method bias (CMB) [49]. Podsakoff and Organ considered that CMB is not serious if the variance explained by a single factor obtained by exploratory factor analysis (EFA) without rotation does not exceed 50% [50]. Tang and Wen’s study pointed out that, based on applications of the test in China, it is generally accepted that the variance explained by a single factor should not exceed 40% [51]. The current study showed that the amount of variance explained by the first factor was 35.13%, which was lower than the critical value of 40%. Therefore, there was no obvious common method bias in this study.

### 3.2. Preliminary Analysis

The result obtained from the preliminary analysis indicated that the average age at which children initially engage with electronic media is 2.45 years (*SD* = 1.40). There is a trend toward younger generations in electronic media usage, suggesting that the younger the child, the earlier they start using such media. In this study, apart from one 2-year-old and one 5-year-old, all children had previous experience with electronic media. Therefore, excluding the 2-year-old cohort (97.1%) and the 5-year-old cohort (99.4%), the proportion of children using electronic media in all other age groups was 100%. Chi-square analysis revealed no significant differences in electronic media usage across various age groups, χ^2^(12) = 10.17, *p* = 0.601.

### 3.3. Descriptive Statistics and Correlation Analysis

The means and standard deviations of the main variables in this study were calculated using descriptive statistical analysis. The details are shown in Table 2.

The associations among the main variables were investigated through Pearson correlation analysis. As shown in Table 2, children’s electronic media use positively correlated with parent–child conflict and internalizing problems. A positive correlation was found between parent–child conflict and internalizing problems. There was a positive association between children’s age and electronic media use. Additionally, children’s gender exhibited negative associations with parent–child conflict and internalizing problems. Given the correlation associated with gender, we treated gender as a control variable in the subsequent analysis.

### 3.4. Moderated Mediation Effect Test

The current study aimed to explore the association between children’s electronic media use and internalizing problems through the mediation of parent–child conflict and the moderating role of age. Hence, we conducted structural equation modeling using the maximum likelihood method in AMOS version 26.0 to assess the hypothetical model. Confidence interval bootstrap tests for bias correction were performed for the model paths, with sampling repetitions set at 5000 and confidence intervals set at 95%. According to Little et al. and Wu and Wen’s suggestions for item packing when items ≥ nine in structural modeling, an isolated parceling should be used for scales containing multiple subscales to pack the individual subscales into a single indicator [52,53]. For unidimensional scales, factorial, correlation, radial, or random algorithms can be adopted. The random method is the most frequently recommended among these packaging methods because it is conceptually independent of established scales and samples [54]. Therefore, we employed the isolated parceling packaging method for the Electronic Media Use Questionnaire with a multidimensional structure, and the random algorithm packaging method for the parent–child conflict scale with a unidimensional structure. Conversely, the internalizing problems scale, despite being a unidimensional scale, was not subjected to packaging due to its limited item count of only seven (<nine items).

A structural equation model was constructed to investigate the relationship between children’s electronic media use and internalization problems, with gender as a control variable. The model fit results were as follows: χ^2^/df = 2.85, RMSEA = 0.05, GFI = 0.97, NFI = 0.96, IFI = 0.97, TFI = 0.96, and CFI = 0.97. The path from children’s electronic media use to internalizing problems was significantly positive (β = 0.47, *p* < 0.001, 95% CI [0.39, 0.55]). In other words, the more children use electronic media, the stronger the likelihood of internalizing problems.

To examine the possible mediating role of parent–child conflict in the association between children’s electronic media use and internalizing problems, we incorporated parent–child conflict into the original structural model. The evaluation of the model fit yielded the following results: χ^2^/df = 2.82, RMSEA = 0.05, GFI = 0.95, NFI = 0.95, IFI = 0.97, TFI = 0.96, and CFI = 0.97. The path from children’s electronic media use to parent–child conflict was significantly positive (β = 0.60, *p* < 0.001, 95% CI [0.53, 0.66]), while the path from parent–child conflict to internalizing problems was also positive (β = 0.58, *p* < 0.001, 95% CI [0.47, 0.68]). The application of bootstrap testing revealed a significant mediating effect (*ab* = 0.13, *p* < 0.001, 95% CI [0.09, 0.18]). These findings indicate that the association between children’s electronic media use and internalizing problems was partially mediated by parent–child conflict, with the mediating effect explaining 73.36% of the overall relationship. In other words, children’s increased use of electronic media was more likely to trigger parent–child conflict and intensified the likelihood of children suffering from internalizing problems.

To further ascertain the moderating role of children’s age in the latter half of the mediated model path, the structural model was expanded by incorporating children’s age as a moderator variable and including an interaction term between children’s age and parent–child conflict (post-centering). The model fit results were as follows: χ^2^/df = 2.94, RMSEA = 0.05, GFI = 0.94, NFI = 0.93, IFI = 0.96, TFI = 0.95, and CFI = 0.96. The specific paths are shown in Figure 2. The results of bootstrap tests showed a significant positive path from the interaction term between children’s age and parent–child conflict to internalizing problems (β = 0.12, *p* < 0.01, 95% CI [0.03, 0.22]), as well as a significant path from parent–child conflict to internalizing problems (β = 0.58, *p* < 0.001, 95% CI [0.47, 0.67]). Consequently, these findings demonstrate that, as children grow older, the association between parent–child conflict and children’s internalizing problems becomes more pronounced.

In addition, we carried out a simple slope analysis to illustrate the moderating role of children’s age in the relationship between parent–child conflict and internalizing problems. The results, as depicted in Figure 3, indicate that parent–child conflict exhibited a significant and positive association with internalizing problems among younger children (β = 0.45, *p* < 0.001, 95% CI [0.32, 0.59]). Furthermore, the positive correlation between parent–child conflict and internalizing problems became more prominent in older children (β = 0.70, *p* < 0.001, 95% CI [0.56, 0.83]). Figure 3 vividly represents the heightened positive correlation between parent–child conflict and internalizing difficulties as children grow older.

## 4. Discussion

### 4.1. Electronic Media Use and Internalizing Problems

This study revealed a positive association between children’s electronic media consumption and internalizing problems, corroborating previous research findings [55] and confirming Hypothesis 1. The social displacement hypothesis posits that if individuals spend too much time using electronic media such as cell phones and computers, they inevitably spend less time communicating in the real world [19]. However, individuals who spend less time socializing in the real world will likely experience diminished interpersonal skills [56] and an increased risk of social anxiety [57,58]. Therefore, the extensive utilization of electronic media by children significantly impinges on their offline communication, leading to a gradual disengagement of children from real-world socialization. This phenomenon can be attributed to the disparities between interactions in the virtual realm and those in reality [59], as well as the predominantly fictitious or surreal nature of the content to which children are exposed in the virtual world [60]. Hence, it is plausible that children who have become accustomed to engaging in online communication may exhibit heightened vulnerability to experiencing discomfort or frustration when engaging in offline communication, potentially exacerbating internalized issues such as anxiety, depression, and social withdrawal behaviors [16,17,18].

Electronic media harbor a plethora of information, yet the nature of the content children are exposed to is not consistently positive. Riddle and Martins conducted a comprehensive analysis spanning two decades which included an assessment of 21 primetime television programs and movies aired on 765 broadcast and cable networks in the United States. Their findings revealed a substantial escalation in the prevalence of violent content [61]. Consequently, as children’s utilization of electronic media increases, the likelihood of their exposure to objectionable online material, such as violence and pornography, increases proportionally [62]. Children’s prolonged exposure to violent content negatively affects their social development [63]. Lin et al. investigated the brain structure of experimental subjects with the help of Voxel-based morphometry (VBM). They found that as an individual’s exposure to online games increased, their hippocampal gray matter density decreased accordingly [64]. As a higher center for learning, the hippocampus is associated with the performance of an individual’s working memory [65]. A child with deficits in working memory is likely to have suboptimal emotional regulation capabilities [66]. Therefore, these impairments significantly impede their ability to adapt effectively to societal demands, exacerbating internalizing difficulties [67].

### 4.2. The Mediating Role of Parent–Child Conflict

The findings also validated Hypothesis 2, demonstrating that parent–child conflict mediated the relationship between children’s utilization of electronic media and internalizing problems. This result reinforces the importance of “technoference” within familial dynamics, as it emerges as a crucial detriment of the quality of family relationships. Specifically, if a family member is addicted to electronic media, their interactions with other family members will be jeopardized, which is not conducive to forming a positive parent–child relationship [68]. Furthermore, it should be noted that the quality of content available on electronic media exhibits variability [69]. Concurrently, the utilization of electronic media by children has been found to affect their sleep patterns and injure their visual health [70]. Therefore, children’s engagement in electronic media tends to prompt parents to adopt restrictive parenting approaches [71], giving rise to conflicts between parents and children [72].

Caution is due here because variations exist in the parental approaches embraced across different countries, driven by diverse cultural contexts [73]. Within Asia, a collectivist cultural orientation prevails, characterizing children as subservient within the family unit. In this framework, stringent discipline is often perceived as beneficial, while its absence is construed as inadequate supervision and care [74]. Conversely, European and South American societies emphasize the significance of treating children with respect and emotional acceptance, associating rigid control with adverse outcomes [75]. Asian parents lean towards cultivating discipline and constraint in child rearing, while South American and European counterparts prioritize tolerance, respect, and acceptance. As a result, parental behaviors concerning children’s exposure to electronic media diverge across cultural landscapes. Yu et al. underscored that Chinese parents exhibited greater involvement in overseeing their children’s electronic media usage than their American counterparts [76]. Similarly, Korean parents accentuated the adverse effects of electronic media on children [77]. However, in the Netherlands, parents assume the roles of supervisors and co-users in their children’s engagement with electronic media, granting older children autonomy in its utilization [78]. Contrasting with Western perspectives, parents in Asian nations such as China and Korea display heightened vigilance and reservations towards electronic media. Moreover, public opinion associates addiction to electronic media with educational failure, leading many parents to view electronic media as a “scourge” [79]. Such extreme perceptions escalate the likelihood of authoritarian parenting and parent–child conflicts [80,81].

It is worth highlighting the notion in attachment theory that individuals who develop insecure attachments with their parents are prone to extending this sentiment to other social groups [26]. Children with insecure attachments have significantly larger late amygdala volumes [82], and the brain-structure change increases children’s responses to adverse stimuli and amplifies the impact of negative events [83]. Hence, the presence of parent–child conflict may have a destructive impact on children’s social and emotional well-being, impeding their ability to establish trust in others. This may manifest in avoiding intimate interactions, heightened feelings of isolation, and an escalation of internalizing issues, such as depression and anxiety [28].

### 4.3. The Moderating Effects of Age

Perhaps the most striking finding of this study is that the pathway of electronic media use through parent–child conflict to internalizing problems is moderated by children’s age, which supports Hypothesis 3. Parent–child conflict was significantly associated with internalizing problems for younger and older children. However, age amplified the positive correlation between parent–child conflict and internalizing problems. Specifically, older children demonstrated increased vulnerability to parent–child conflict, leading to a more pronounced expression of internalizing problems than in their younger counterparts.

The concept of bidirectionality in parent–child relationships highlights the involvement of reciprocal influence in the social dynamic between parents and children [84]. This implies that children are not merely passive recipients in the parent–child relationship, but rather that their age and temperament have counteractive effects on this relationship [85]. As children mature, they progressively expand their social horizons and develop greater autonomy [86]. Therefore, when parents persist in perceiving their children to be immature and exhibit excessive concern over them, akin to their behaviors during their children’s early years, the child tends to interpret such behavior as an interference or as a negation of their autonomy [87]. This disrespectful approach is prone to inciting emotional difficulties in the child [88].

Additionally, as children grow up, they progressively encounter intensified academic demands and interpersonal rivalries [89]. Compared to students in other nations, students in some Asian countries devote much time to their educational pursuits and frequently experience severe psychological strain due to excessive after-school programs [90]. The prevailing adherence to Confucianism in China is pervasive, leading to the adoption of examination-based talent selection methods [91]. Consequently, Chinese parents commonly hold elevated expectations for their children’s educational achievements, which might place a heavy psychological burden on older children [92]. Evidence suggests that psychological stress or depression generally leads to heightened susceptibility to negative events and intensified consequences of adverse events [93]. Therefore, compared to younger children, when older children are confronted with stressful situations, they may experience parent–child conflict more acutely and are more likely to have internalizing problems.

### 4.4. Limitations and Suggestions for Future Research

There are several noteworthy limitations inherent to the present study. Firstly, it must be acknowledged that this study did not employ a causal experimental or a longitudinal follow-up design. Consequently, it was challenging to effectively investigate the causal associations among children’s utilization of electronic media, parent–child conflict, age, and internalizing problems. In other words, the model used might not be the only model supported by the data. Hence, it is recommended that future research endeavors embrace an experimental or longitudinal design to extensively elucidate the underlying effects mechanism among the variables under investigation.

Secondly, implementing the convenience sampling method for data collection in this study might warrant scrutiny. Thus, the generalizability of the findings may be exclusively constrained to the northern region of China and to those children who appear to be using electronic media. Moreover, parents from different cultural backgrounds may adopt varied parenting styles when addressing children’s use of electronic media, which could further influence the manifestation of parent–child conflicts. Therefore, subsequent research should utilize a broader and more representative sample from diverse ethnic groups, regions, and cultures to thoroughly assess the association between children’s electronic media use and its outcomes.

Thirdly, the data for this study were sourced from parents residing with their children, predicated on the assumption that such parents possess significant insight into their children’s behavior and experiences. However, some working parents might need more time to monitor their children closely. We should have inquired about the parents’ parenting styles, the daily amount of time they spend with their children, or their confidence in evaluating their child’s behavior, which might affect the accuracy of our study’s results. Thus, we will enhance our understanding of these factors in future research to yield more convincing results. 

Fourthly, this study examined children’s internalizing problems across a broader age range and found no significant correlation between age and internalizing problems. However, due to the utilization of a unidimensional measurement scale for internalizing problems in our study, it could not capture the qualitative differences in various aspects of internalizing problems (such as depression, anxiety, and social withdrawal) that may evolve with age. Therefore, future research could employ alternative tools that comprehensively depict internalizing problems, aiming to provide a more nuanced understanding of how these issues transform with age. 

Lastly, it is essential to note that the data relied solely on parental reports, thus somewhat neglecting the “voices” and perspectives of the children. Subsequent studies should incorporate children’s perspectives to systematically explore the relationships among children’s age, electronic media use, parent–child conflict, and internalizing problems, accounting for the diverse viewpoints of both parents and children.

## 5. Conclusions

In summary, the results of this study revealed a noteworthy and affirmative correlation between children’s utilization of electronic media and the display of internalizing problems. The involvement of parent–child conflict was identified as a mediator in this connection. Furthermore, the age of the children was observed to moderate the relationship between parent–child conflict and internalizing problems. Specifically, the positive association between parent–child conflict and internalizing problems intensified with the children’s increasing age.

## Figures and Tables

**Figure 1 behavsci-13-00694-f001:**
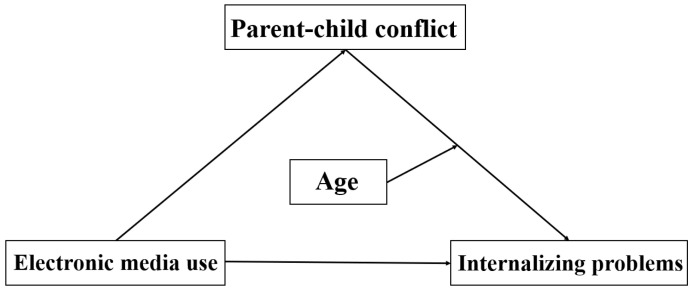
Conceptual framework for hypothesis testing.

**Figure 2 behavsci-13-00694-f002:**
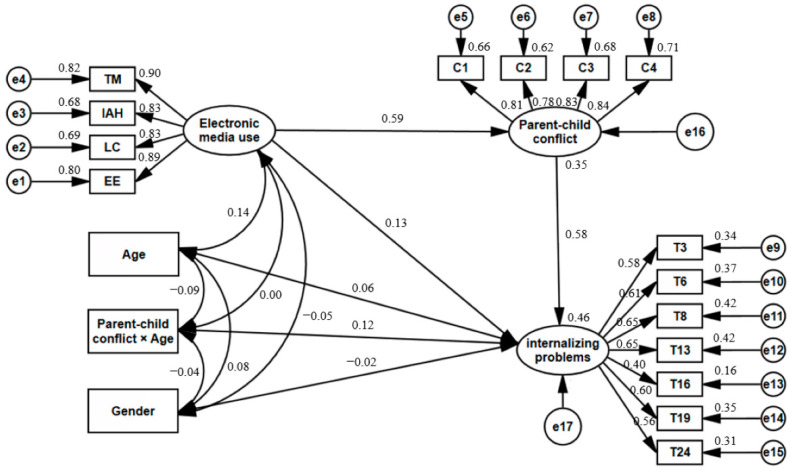
Mediating effect of parent–child conflict on the relationship between electronic media use and internalizing problems and the moderating role of children’s age.

**Figure 3 behavsci-13-00694-f003:**
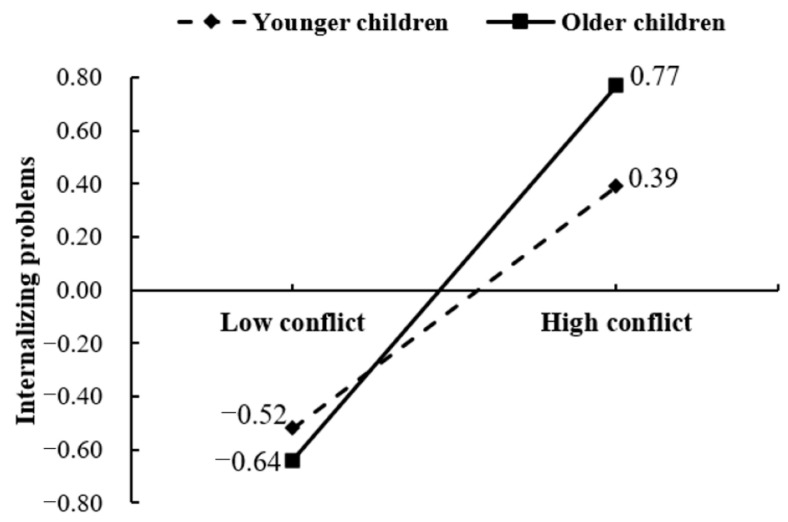
Simple slope analysis of the interaction between parent–child conflict and children’s age in internalizing problems.

**Table 1 behavsci-13-00694-t001:** Demographic characteristics of the parents.

	Father	Mother
Mean age	34.10 years (*SD* = 5.11)	32.34 years (*SD* = 4.62)
Education (%)		
Primary and below	0.4%	0.6%
Junior high	2.4%	2.2%
Senior high and middle school	9.3%	8.6%
College and bachelor’s degree	70.6%	78.4%
Postgraduate and above	17.4%	10.3%
Occupation (%)		
Managers	25.5%	18.5%
Professionals and technicians	20.1%	15.3%
Clerical support workers	8.6%	9.1%
Service and sales workers	11.5%	13.7%
Skilled agricultural, forestry, and fishery workers	1.6%	0.6%
Craft and related trades workers	2.6%	1.0%
Armed forces occupations	2.2%	0.3%
Full-time parents	0.6%	13.4%
Unemployed	0.9%	1.9%
Freelancers	26.3%	26.3%
Monthly income (%)		
≤2999 CNY	1.9%	12.8%
3000–5999 CNY	15.0%	25.9%
6000–8999 CNY	29.6%	32.6%
9000–11,999 CNY	25.9%	17.7%
≥12,000 CNY	27.6%	11.0%

**Table 2 behavsci-13-00694-t002:** Descriptive statistics and correlations of the main variables.

Variables	*M*	*SD*	1	2	3	4	5
1. Gender	0.52	0.50	—				
2. Age	5.23	1.91	—	1			
3. Electronic media use	33.19	10.68	−0.05	0.15 **	1		
4. Parent–child conflict	25.97	8.29	−0.09 *	0.04	0.53 **	1	
5. Internalizing problems	2.66	2.53	−0.08 *	0.07	0.42 **	0.54 **	1

Notes: Gender: 0 = boy, 1 = girl; * *p* < 0.05, ** *p* < 0.01.

## Data Availability

The data that support the findings of this study are available from the corresponding author upon reasonable request.

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
