# Peer review of "Association between Electronic Media Use and Internalizing Problems: The Mediating Effect of Parent–Child Conflict and Moderating Effect of Children’s Age"

_behavsci, 2023, doi:10.3390/bs13080694_

Round 1

Reviewer 1 Report

1.  Line 41  It is not clear how the issue of grandparents suddenly appears here; I had thought the paper was about parents and their children.  If this is correct and retained, it needs more explanation.

2.  Line 216  Table 1.  At line 189 it is stated that the scale has 21 items and is scored on a three point scale, possibly indicating scores from 21 to 63.  However, the mean score is shown as 0.46 in Table 1.  The way the three key variables were scaled needs to be explained more clearly so the means reported make more sense.

3.  Line 216, Table 1.  With an average age of 5.23 and some children much younger, I think more clarification is needed.  Was the focal child the only child in the family?  Perhaps, the focal child was age 2 but the parents had another child age 10 such that the older child was using electronic media but not the two year old child.  It might be nice to know how the percentage of children using electronic media changed with their age.  If the child isn't using electronic media and, thus, one might expect low parent-child conflict, the role of age might be confounding, for younger children not using media, creating a sink hole of low age, zero use, zero conflict, which could distort regression coefficients.  Should the study sample be limited to those children who appear to be using electronic media - or was this already done?  Please clarify. 

4.  The role of causality may need more discussion.  Might not parent-child conflict lead children to engage more with electronic media as a way of escaping that conflict?  Having internalizing problems might lead to more electronic media use?  If a child is depressed and withdrawn might that lead to more parent-child conflict?  The model seems supported by the data in Figure 2, but that doesn't mean that the model used is the only one that the data might support. 

Reviewer 2 Report

The manuscript entitled “Impact of Electronic Media Use on Internalizing Problems: The Mediating Effect of Parent–Child Conflict and Moderating Effect of Children’s Age” is an interesting and valuable contribution to the relationship between the use of electronic media and psychological difficulties. This correlational study relies on a questionnaire distributed to a sample of parents. In my modest opinion, the manuscript deserves publication after a few concerns are adequately addressed by the authors. The following are these concerns.

 First and foremost, the study is an instance of correlational research. The authors are justified in labeling some factors as “predictors” or in referring to such factors as “contributors”. However, I would recommend that they abstain from mentioning their “impact”. A correlation is not an instance of a cause-effect relationship examined within an experiment.

The introductory section would benefit from a broader and more in-depth review of the large diversity of “internalizing problems” that may exist among children of different ages. I would also focus on a clear rationale for the selection of the particular age range that the authors report in their manuscript. To this end, consider that in the method section, the authors state that “[t]he age range of the respondents’ children was 1.08 to 13.09 years with a mean age of 5.23 years (SD = 1.91)”. This substantial age range gives children’s age enough variability to be useful in the analyses that the authors have chosen for hypothesis testing. Yet, statistical considerations are different from theoretical ones. That is, the authors may need to consider that psychological difficulties (including internalizing problems) may be qualitatively different as the age of a child increases.

Does the questionnaire provide parents with an operational definition of “electronic media”? Different parents may not share the same understanding of the concept.

The assumption upon which the questionnaire is built is that parents know what their children do daily. However, if they have a full-time job, caregivers may be more informed. Did the authors attempt to gather information about the amount of time on average parents spend with their children daily? Irrespective of work responsibilities, some parents may be more attentive to their children’s behavior than others. Why were not parenting styles examined? What do we know about the parents’ demographic characteristics? At the very minimum, why didn’t the authors allow parents to indicate the confidence with which they evaluated their children’s behavior?

In the discussion of the results, cultural diversity in the extant literature may be considered more broadly. Namely, what is the evidence regarding cultural differences in parenting styles that may account for the authors’ results? What is the extent to which their results generalize to other populations of parents around the world?

Minor editing of the English language is required.

Round 2

Reviewer 1 Report

Excellent work for the revisions!